# External Focus or Differential Learning: Is There an Additive Effect on Learning a Futsal Goal Kick?

**DOI:** 10.3390/ijerph19010317

**Published:** 2021-12-29

**Authors:** Sara Oftadeh, Abbas Bahram, Rasoul Yaali, Farhad Ghadiri, Wolfgang I. Schöllhorn

**Affiliations:** 1Faculty of Physical Education and Sport Sciences, Kharazmi University of Tehran, Tehran 14155-6455, Iran; bahram@khu.ac.ir (A.B.); r.yaali@khu.ac.ir (R.Y.); ghadiri@khu.ac.ir (F.G.); 2Department for Training and Movement Science, Johannes Gutenberg-University Mainz, 55128 Mainz, Germany

**Keywords:** differential learning, OPTIMAL theory, external focus, motor learning, futsal

## Abstract

(1) Background: How to optimally promote the process of acquiring and learning a new motor skill is still one of the fundamental questions often raised in training and movement science, rehabilitation, and physical education. This study is aimed at investigating the effects of differential learning (DL) and the elements of OPTIMAL theory on learning a goal-kicking skill in futsal, especially under the conditions of external and internal foci. (2) Methods: A total of 40 female beginners were randomly assigned to, and equally distributed among, five different interventions. Within a pretest and post-test design, with retention and transfer tests, participants practiced for 12 weeks, involving two 20-min sessions per week. The tests involved a kicking skill test. Data were analyzed with a one-way ANOVA. (3) Results: Statistically significant differences with large effect sizes were found between differential learning (DL) with an external focus, DL with an internal focus, DL with no focus, traditional training with an external focus, and traditional training with control groups in the post-, retention, and transfer tests. (4) Conclusions: The results indicate the clear advantages of DL. It is well worth putting further efforts into investigating a more differentiated application of instructions combined with exercises for DL.

## 1. Introduction

How to optimally promote the process of acquiring and learning a new motor skill more effectively is still reckoned as one of the core questions in movement science, therapy, and training science [1]. Classically, the traditional approach to skill acquisition is based on reductionist-atomistic thinking, accompanied by an excessive emphasis on cognition since the “cognitive turn” in the 1960s [2]. This is a way of thinking that is mainly based on linear models, explicit verbalization, a lot of imitations, and the internalization of knowledge about a correct prototype that is shown by a lot of repetition and corrective instructions to avoid making mistakes [3]. In the same context, learners are expected to copy the role models of motor patterns [4], mostly derived from the averages of the best athletes in their disciplines. To support the learners during this process, manual [5], mechanical [6,7,8], verbal [9], and visual [10] guidance is also suggested in the form of augmented feedback [11,12,13]. Following this traditional logic of a control loop with external feedback, the main theoretically assumed key factors influencing skill learning [14] would first need to be identified, and their interactions would need to be appropriately understood by practitioners and coaches to establish a strong theoretical basis for an appropriate pedagogical approach [15].

More recently, the assumptions proposed for this traditional repetitive and model-oriented approach have been increasingly challenged by the upcoming system dynamics and complexity theory in general [16], and football in particular [17]. From a practical perspective, the repetitive approach became questioned. The process of acquiring more difficult gross motor movements, in which all learners typically begin with the same exercise and then progress to the final learning goal with the identical sequence of exercises, is increasingly being challenged. In this process, each exercise must be repeated until only small differences are made apparent from the partial target model. When the purported target movement has been successfully achieved, the approach stagnates in the exclusive repetition of the movements, and only the hope for additional conditional progress provides further motivation. By reliably identifying individuals not only through their biomechanical movement patterns [18,19], but also through a recognition of the emotions [20] and the fatigue phases [21] within an individual, as well as of individuality, even across disciplines [22], the main premises of the traditional approach have been deconstructed. As a result, the question arises about the necessity of repetition and prototypes, since individuals, situations, and ongoing changes must be dealt with. Given the extremely low probability of executing two identical movements [21,23,24], particularly during a futsal match or similar games with an infinite number of player constellations, traditional learning methods rely primarily on repetitions to realize an ideal movement, and sustainability in error corrections should be reconsidered [25].

Although differential learning (DL) theory is explicitly silent about psychological aspects and applies instructions with both internal (joint angles) [26] and external foci (variable targets) [27], as well as with a metaphoric (moving in animal styles) [28] focus in its experiments, DL is occasionally associated with the internal focus [29]. In the case of learning a tennis forehand stroke [30], the focus should be on performing the forehand with an extended elbow, and then with a flexed elbow, then stiff knees, and so on [31]. In a volleyball experiment, the DL idea of increased noise was realized by means of moving similar various animals (metaphorically) that were creatively announced by each server [28]. To achieve maximum variety in field hockey, participants either changed their bodily positions in response to some internally connected instructions, or they had to change the target for which they were externally attempting [27]. Nonetheless, systematic investigations concerning the interdependencies of variable movements and the foci of instruction are missing.

A quite different and more recently suggested approach to support motor learning has been developed with the OPTIMAL approach [32]. While the DL approach, in the beginning, focused more on the variety of external stimuli that result from an interaction of active and passive forces, as well as on the conditions applied on the organization of the central nervous structures through variable movements via mechanoreceptors [16,33] before the increased fluctuations were adapted to an individual’s movement noise [26] the OPTIMAL approach is more rooted in sports psychology, as it addresses the specific mental aspects of movements, such as motivation and attention [34]. Lewthwaite and Wulf proposed the OPTIMAL theory of motor learning, involving two motivational variables (enhanced expectancies and support for autonomy) and an attention variable (e.g., the focus of external attention) affecting optimal performance. The enhanced expectancies, mainly initiated by corresponding instructions, refer to an increase in individual expectations for positive experiences or success [34], while support for autonomy is said to be a condition that supports the need of people to be in control or to be autonomous over their actions [35]. As for the attention variable, studies on the OPTIMAL theory indicate that having an instructed external focus of attention on the intended movement effect (e.g., implement trajectory, hitting the target, exerting force against the ground) typically results in more effective and efficient performances or learning, as compared with the internal focus on body landmarks [36]. In fact, OPTIMAL theory predicts that an external focus of attention is more beneficial for motor learning than an internal one [37].

Therefore, the general aim of this study was to evaluate the effects of the DL approach and the elements of the OPTIMAL theory on the goal-kicking performance in futsal, in comparison to a purely repetitive approach. More specifically, the hypotheses to be tested were that the DL intervention would not only outperform the external-focus intervention, but also that both interventions would have significantly larger effects compared to repetition-only learning, and, finally, that the combination of both interventions would outperform the DL-only intervention, with or without an internal attentional focus [29,36].

## 2. Materials and Methods

### 2.1. Participants

A total of 40 school girls, aged 16 to 18 years and recruited from the city of Baghmalek, Khuzestan Province, Iran, voluntarily participated in this study, and were randomly assigned to five equal groups with different instructions during the interventions: (a) DL with an external focus (DL/EF n = 8); (b) DL with an internal focus (DL/IF n = 8); (c) DL without a specific focus (DL/C n = 8); (d) Traditional training with an external focus (T/EF n = 8); and (e) Traditional training without a specific focus (T/C n = 8). The inclusion criteria were as follows: no history of formal futsal training, and no reported disease or injury. The exclusion criterion is specified as follows: absenteeism in any of the training sessions. The study was approved by the university’s Institutional Review Board, with the ethics code: “IR.KHU.REC.1399.04”. Using G-Power (version 3.1), the sample size was estimated at the significance level of 0.05, with a 0.80 statistical power, and an effect size of 0.6 (medium to large effect size), using the statistical one-way ANOVA method [37,38]. Accordingly, 40 people were assigned to five groups with different interventions.

The demographic information for all the groups is presented in Table 1. Specifically, the average age of all the participants was 16.97 ± 2.28 years, their mean body mass was 58.62 ± 4.27 kg, and their mean body height was 159.36 ± 3.73 cm.

Having chosen the participants according to the inclusion criteria, the consent form was completed, and the COVID-19 protocols were observed for all the people who played a role in the study, including the subjects, their parents, and the examiners. All the health protocols were applied to all of the individuals, according to the COVID-19 requirements. All procedures and measurements complied with the Declaration of Helsinki (54th Revision 2008, Korea) regarding human subjects.

### 2.2. Design

A pretest and post-test design, with retention and transfer tests, was chosen for this study. Then, a futsal shooting skill was evaluated as an example of learning a gross motor skill. The chronological schedule of the design is shown in Table 2. After the pretest, all the subjects participated in a 12-week intervention period, with two 20-min sessions per week. Ten minutes after the last intervention, a post-test was conducted. One week after the post-test, all the subjects participated in the retention and transfer tests.

### 2.3. Tests

All tests and interventions were executed applying a standard futsal ball and a standard futsal goal in a standard indoor futsal court.

In the goal-shooting test, participants had to shoot the ball from a 6-m line towards the goal, without a goalkeeper, in seven different situations, with each one repeated five times. Overall, each participant performed 35 shooting movements in a blocked order (7 situations × 5 times = 35 trials). We used the average scores obtained in the 35 trials as the final score of each person in the shooting test. The goal-shooting test was adopted from Schöllhorn [26,39]. The seven different goal-shooting situations were as follows:Five immobile futsal balls were shot towards the goal after a short approach from Position 1 (Figure 1);Five futsal balls were shot towards the goal after 10 m of dribbling from Position 1 (Figure 1);Five futsal balls were shot towards the goal after 5 m of dribbling from Position 2 (Figure 1);Five futsal balls were shot towards the goal from Position 1 after a pass from the right (Figure 1);Five futsal balls were shot towards the goal after 5 m of dribbling from Position 3 (Figure 1);Five futsal balls were shot towards the goal from Position 1 after a pass from the left (Figure 1);Five futsal balls were shot towards the goal from Position 1 after crossing a 40-cm height obstacle with a vertical jump (Figure 1).

The participants’ shooting positions, and the ways in which they scored their shots, are shown in Figure 1.

The precision of the shots was measured by dividing the goal into scoring zones. The zones were chosen based on the likelihood that a goal would be scored in each one of them. Meanwhile, because of this, the regions that were more difficult to reach by a goalkeeper were scored higher, and vice versa. Shots that closely missed the goal still scored 1 point (Figure 2) [39]. Additional spectators were asked to produce a more stressful environment, and to monitor the learned content for stability against psychological disturbances during the transfer test. According to the findings by Henz et al. [31,40], differential learning is accompanied by a downregulation of the frontal lobe that is associated with higher stress resistance, and it should consequently lead to better test performances in the DL intervention groups.

### 2.4. Interventions

The DL exercises are described in Table 3. Each exercise was assigned a number. The numbers were written on slips of paper and were all put in a box. The slips of paper with the numbers were then randomly drawn from the box during each intervention session. The assigned exercises were then performed by the participants. Up to three of the exercises listed in Table 3 were combined.

Depending on the focus of each group, the participants received the following instructions, with respect to their focus of attention during the kicking movement. The specific instructions were specified as follows:

DL/EF Practices: Firstly, the DL/EF practice group received verbal instructions for the movement to be executed according to the combinations chosen from Table 3, and secondly, they received a verbal instruction for the external focus, where they had to focus on a target related to the area of the goal they intended to shoot at (e.g., shooting the ball with the middle of the foot while focusing on the upper left corner of the goal).

DL/IF Practices: The DL/IF practice group received, firstly, verbal instructions for the movement that had to be executed according to the combinations chosen from Table 3, and secondly, a verbal instruction for the internal focus, aimed at focusing on a body-related zone (e.g., shooting the ball with the midfoot, and focusing on the moment the foot hits the ball.)

DL/C Practices: The DL/C practice group received only verbal instructions for the movement that had to be executed according to the combinations chosen from Table 3 (e.g., shooting the ball with the middle of the foot).

T/EF Practices: The T/EF practice group received a visual nonverbal demonstration for the most common futsal shooting pattern provided by the teacher. Then, the participants were asked to follow the prototype pattern provided, with an additional instruction for an external focus of attention (e.g., shooting the ball with the toes and focusing on the upper left corner of the goal).

T/C Practices: The T/C practice group only received a visual nonverbal demonstration for the most common futsal shooting pattern, provided by the teacher to compellingly be imitated (e.g., shooting at the goal with your toes).

All the practice sessions and instructions were administered by one of the authors of this article, who is also a futsal expert.

### 2.5. Statistical Analysis

The acquisition and learning rates were assessed using a one-way ANOVA test. The variables were tested to assess the normality (Shapiro–Wilk Test), homoscedasticity (Levene’s test), and sphericity (Mauchly’s test) (*p* > 0.05). The effect sizes (r or η²) were also estimated for all of the comparisons [43], and they were comparatively classified as a small effect (r = 0.10 ~ d = 0.2 or η² = 0.01), a medium effect (r = 0.25 ~ d = 0.5 or η² = 0.06), or a large effect ((r = 0.37 ~ d = 0.8 or η² = 0.14) [44]. The level of significance was set to 0.05, and the statistical analyses were conducted using SPSS 23.0 (IBM Corp., Armonk, NY, USA).

## 3. Results

The descriptive results obtained after all four tests were conducted on all five groups are graphically displayed in Figure 3. Specifically, the highest acquisition rates were achieved by the differential training groups. Within the DL groups, the group with an external focus achieved the highest scores in the post-test and in the retention and transfer tests.

The one-way ANOVA test showed that there was no difference between the subjects in the pretest, but for the post-test, the data showed a statistically significant difference between the groups with a large effect size (F_(35.4)_ = 698.55, *p* = 0.001, η^2^ = 0.98). Moreover, the results of the LSD post hoc test revealed a statistically significant difference between the DL/EF, DL/IF, and DL/C groups compared to the T/EF and T/C groups (*p* < 0.05; r > 0.83), with the external, internal, control differential, external, and control traditional groups accounting for the highest scores, followed by the control traditional group. Moreover, an ANOVA of the retention test data indicated a statistically significant difference between the groups (F_(35.4)_ = 2104.9, *p* = 0.001, η^2^ = 0.99). The LSD post hoc test results show a statistically significant difference between the DL/EF, DL/IF, and DL/C groups and the T/EF and T/C groups (*p* < 0.05; r > 0.82), and the highest scores indicate the DL groups with external and internal foci, as well as the control group with no focus instructions, followed by the traditionally repetitive external and control groups (Table 4).

The ANOVA of the transfer test revealed a statistically significant difference between the groups (F_(35.4)_ = 739.28, *p* = 0.001, η^2^ = 0.98). Furthermore, the LSD post hoc test results show a statistically significant difference between the DL/EF, DL/IF, and DL/C groups and the T/EF and T/C groups (*p* < 0.05; r > 0.74), and the highest scores related to the external, internal, control differential, external, and control traditional groups (Table 4). Most intriguingly, all the effect sizes were large in both the ANOVA and in all of the post hoc tests, on the basis of Cohen’s classification [44].

## 4. Discussion

The purpose of this study was to explore the distinguished effects of various forms of the differential learning approach [16], combined with elements of OPTIMAL theory [32], on learning the futsal goal-shooting skill. The statistical analysis revealed significant differences between the interventions DL/EF, DL/IF, DL/C, T/EF, and T/C training in the post-, retention, and transfer tests. The present test results were consistent with previous research conducted on DL (e.g., [45,46,47]), which show the advantages of DL in comparison to the rather repetitive approaches in the acquisition and learning phases. However, gender must be kept in mind as a possible moderator, as other studies have examined students of the same age, but only male students. 

A 12-week intervention was used in this study, as opposed to the 4-to-6-week interventions used in earlier DL-related football studies [26,39]. Compared to repetition-based training, it appears that the benefits of differential learning increase over time. This has also been observed in research on tennis, with varying intervention durations, where the rising advantage of DL increased over time [48,49,50]. As the intervention duration increases, the DL advantages are mainly gained over traditional training in the acquisition phase, and they remain for the learning phase. These results are also consistent with the assumption that, in the DL approach, fluctuations in learning subsystems are advantageously used for learning by destabilizing the system to prepare a self-organized phase transition [45]. By amplifying such fluctuations, the system can potentially experience new solutions, involving new combinations of given activations [19]. As a result, a self-organizing process is initiated and exploited that stimulates the system to develop a new coordination strategy that typically results in more effective or stable patterns of movement [45]. The amplified fluctuations and intermittencies tend to increase the variances in other anatomical regions of the body, and lead to a highly unpredictable adaptation process [45]. The whole process finally corresponds to the somehow unspecific formulated cybernetic law of requisite variety [51], or to the “bliss of abundance” [52,53]. The first was formulated with “only variety can destroy variety”, in general, as a law in cybernetics for the regulation of processes in nature and can be observed impressively in this study and the studies connected with DL: The additional variation in exercise caused the initial large variation in the hit performance to “destroy” it, resulting in a greater hit performance. Gelfand and Latash [52,53] formulated something similar, but more poetically, with the “bliss of abundance” as an alternative view of movement variety that occurs even in repetitive motor learning processes, and thus plead for a rethinking of the traditional motor control theories that are mainly based on singular solutions.

In contrast to traditional prototypical training, DL exercises encourage learners to actively perform movement errors (e.g., throwing to the left instead of forwarding), rather than avoid them [54]. Participants performed shooting to the left, to the right, upwards, and shooting straight. Errors are considered as neutral fluctuations (“you never know, what they are good for”) and are not only experienced to know what is correct, which was already included in contrast learning and would give the learner convergent guidance in contrast to divergent self-organization. In DL the fluctuations are essential elements for a more stable training of the neural nets involved. 

Our findings are consistent with those of other researchers [55,56,57] who have examined the typology of attention focus, which, according to the OPTIMAL theory, suggests that an external focus of attention results in increased learning skills when compared to an internal focus of attention. Nevertheless, based on the results, the focus of external attention combined with DL, as in the DL/EF condition, resulted in a higher, statistically significant improvement than the T/EF. These results provide an indication of the stronger influence of the physically dominated variations of the DL approach, compared to the mentally supportive traditional training method. Regarding the futsal goal-shooting transfer test, statistically, significant differences were observed between all five intervention groups.

A neurophysiological explanation for these phenomena would be appealing but has, so far, been problematic. This is because the available EEG studies on both DL and external foci always refer to acute effects, but not to medium-term effects, after multiple interventions. This is complicated by the fact that the EEG studies on DL focus on the effects immediately after a whole series of movements, whereas the studies on an external focus tend to focus on the EEG activation immediately before and during a single movement. The EEG studies on DL show an increase in power primarily in the frontal brain areas in the alpha and theta frequency bands after the intervention [31,40] whereas a reduction in these frequency bands in the central brain areas is observed during external focus tasks [58]. A decrease in alpha power was also observed during a dart-throwing task with an external focus, in comparison to one with an internal focus [59], but, unfortunately, the theta band was not analyzed, which is assumed to be of major interest for high-concentration tasks [60]. The degree to which these phenomena depend on the complexity of the movement task, the performance level, or the age of the subjects requires further investigation [61,62]. The same applies to the question of the extent to which a decrease in the alpha frequencies during training leads to an increase in the alpha frequencies after training, or the extent to which a sustained increase during sleep leads to a permanent change. Nonetheless, the practical interventions derived from OPTIMAL theory seem to be a complement to DL that could be of great interest when it comes to the absolute limits of performance.

The current study findings were based on two assumptions. First, that DL exercises lead to detecting adaptive solutions through increased fluctuations in the individual, and they appear to support learning ability [46]. Second, that DL exercise, using an optimal relation of the internal and external noise frequencies [63], moves the brain to a more stress-resilient state by downregulating frontal lobe activation [31]. The participants in all the DL-related groups were able to establish an adaptive response to the task modification, even though we included spectators in the test and created a more stressful environment.

Epistemologically, the applied statistics and methodology do not claim to be generalizable. Therefore, what is often interpreted as a limitation turns out, upon closer inspection, to be an aspect to be considered in future studies. One such aspect is related to the influence of the sex of the subjects. While this study was conducted on pubescent girls, it would be of interest to perform the same study with boys or adults. Moreover, in this research, long-term follow-ups were not conducted, which should be used in future research. Given that the present research was performed on beginners in a certain culture, future research should be performed on skilled individuals with different cultural backgrounds as well. In future research, other sports skills, including team behavior, should also be considered. In this study, the highly variable intervention method of differential learning, with assumingly chaotic characteristics, was used to teach a single futsal skill. In the search for optimal variation structures, future studies should also differentiate gradual and chaotic differential learning approaches by adapting the noise of the exercises to the noise inherent to the learner to find the optimal stochastic resonance [49], as well as their interactions with internal and external foci. Another interesting question is the influence of the test sequence. Future research would have to clarify to what extent the interventions have different influences on blocked or random test orders.

## 5. Conclusions

Since all the DL interventions, whether with external or internal foci, showed better performance in the post-, retention, and transfer tests than all combinations of traditional training with the foci, the results suggest the more beneficial influence of the DL training method compared to the traditional training methods, and compared to the focus interventions. Nonetheless, attentional foci seem to provide a kind of positive fine-tuning in addition to DL intervention. It would be of great interest to know what additional influences the other aspects of OPTIMAL theory have in connection with DL, or whether these are possibly formed and promoted precisely by DL.

The DL and the OPTIMAL approaches are both still in the beginning stages of their research course development. Thus, what has been obtained from the present study can be added as further evidence to the body of theoretical basics in these theories. To enhance and create more effective exercises for the acquisition of motor skills, research findings should be made available to instructional and learning designers [64].

## Figures and Tables

**Figure 1 ijerph-19-00317-f001:**
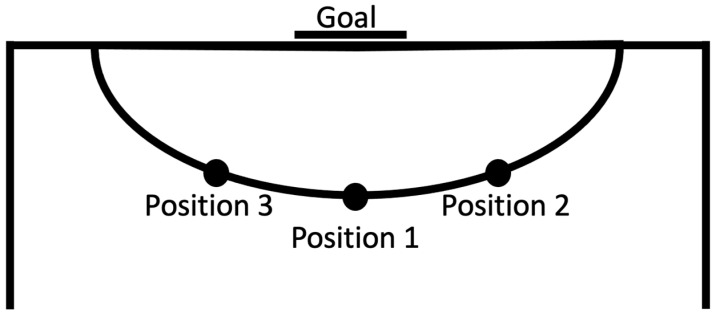
Participant positions for goal-shooting.

**Figure 2 ijerph-19-00317-f002:**
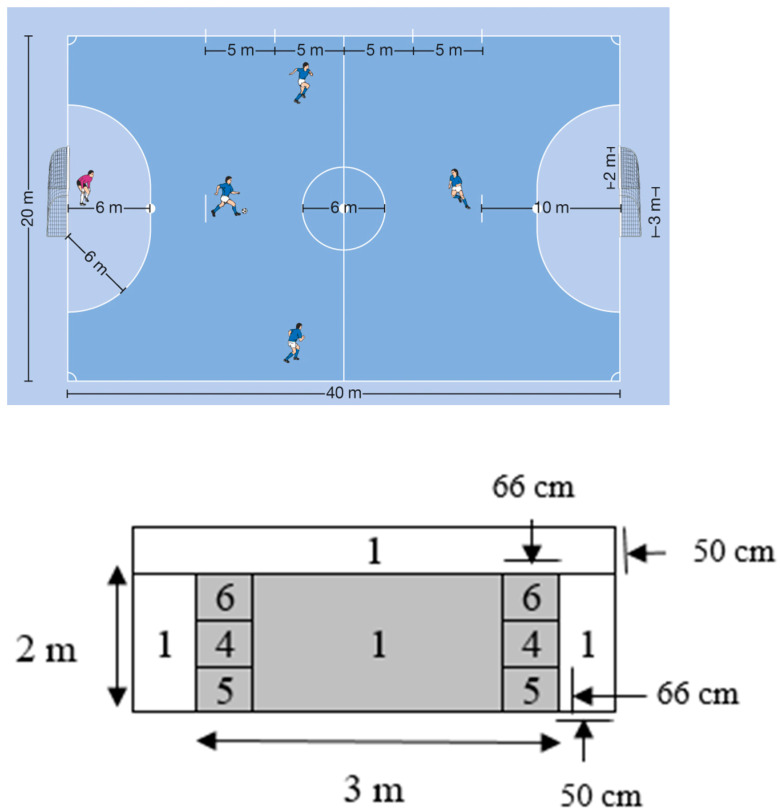
The goal-shooting test scores (Schöllhorn et al., 2012).

**Figure 3 ijerph-19-00317-f003:**
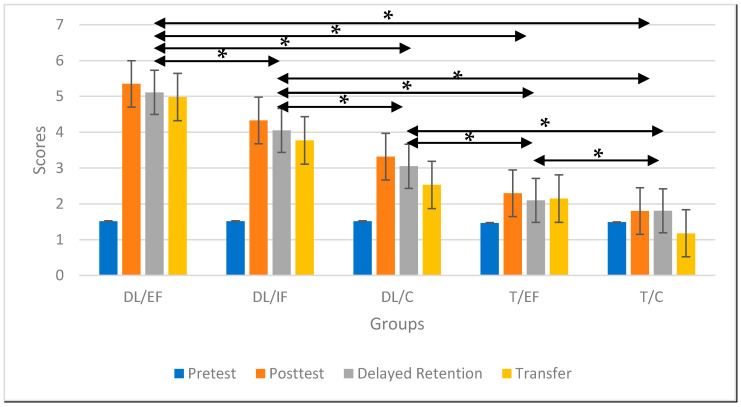
Average scores and standard deviations of pre-, post-, retention and transfer test results for the five intervention groups. DL/EF: differential learning with an external focus; DL/IF: differential learning with an internal focus; DL/C: differential learning without a specific focus; T/EF: traditional repetitive training with an external focus; T/C: traditional repetitive training without a specific focus. * Major statistically significant differences between groups (in all tests, a significant difference was observed between all groups).

**Table 1 ijerph-19-00317-t001:** Demographic characteristics of the five intervention groups.

Variable	Total Mean ± SD	DL/EF Mean ± SD	DL/IF Mean ± SD	DL/C Mean ± SD	T/EF Mean ± SD	T/C Mean ± SD
Age (year)	16.97 ± 2.28	17.3 ± 2.27	17.07 ± 2.16	16.87 ± 2.11	16.79 ± 2.33	16.82 ± 2.53
Mass (kg)	58.62± 4.27	59.74 ± 4.09	58.83 ± 4.47	58.74 ± 3.96	58.4 ± 4.16	57.41 ± 4.67
Height (cm)	159.36 ± 3.73	159.28 ± 3.22	159.99 ± 3.92	158.33 ± 3.72	160.12 ± 4.09	159.11 ± 3.74
Number	40	8	8	8	8	8

**Table 2 ijerph-19-00317-t002:** Study design.

Groups	Pretest	Training	Post-Test	Retention Test	Transfer Test
**DL/EF**	Before the start of the training period, a futsal shoot pretest had been taken by all the groups.	12-week DL exercises (two 20-min sessions per week) with external focus	Ten minutes after the end of the last training session, the futsal shooting test was taken.	One week after the post-test, all the groups performed a futsal shooting test.	One week after the post-test, the futsal shoot transfer test (in the presence of spectators), immediately after the retention test was performed.
**DL/IF**		12-week DL exercises (two 20-min sessions per week) with internal focus			
**DL/C**		12-week DL exercises (two 20-min sessions per week) with no attention instruction			
**T/EF**		12-week traditional exercises (two 20-min sessions per week) with external focus			
**T/C**		12-week traditional exercises (two 20-min sessions per week) with no attention instruction			

**Table 3 ijerph-19-00317-t003:** DL Exercises [41,42].

Standing position	Smooth and vertical, bend forward and bend backward
Joint position	The maximal fixed, the middle position, and the maximal stretched
Hand of the standing foot side	Overhead, under the hip, in front, back, and lateral
Joint movement	Flexion, extention, abduction, adduction, internal and external rotation
Standing foot	Standing on toes, standing on heels, and standing on the whole feet
Shooting direction	Shooting to the center, the left, the right, the center, the top, and the bottom
Movement velocity	Slow, submaximal, and maximal
Hand of the shooting leg side	Overhead, under the hip, in front, back, and lateral
Eyes	Both eyes open, both eyes closed, left eye closed, and right eye closed
Foot position	Left foot front, right foot front, and feet parallel
Ball	Large, small, heavy, light balls, and other sports balls
Muscles	Maximally tensed, activated, and relaxed

**Table 4 ijerph-19-00317-t004:** Fisher’s least significant differences (LSDs) of averages comparing changes in pre- and post- retention and transfer tests. * Deemed to be statistically highly significant, as observed.

		Prepost Test			Preretention Test			Pretransfer Test		
(I) Group	(J) Group	Mean Difference (I-J)	SD	ES (r)	Mean Difference (I-J)	SD	ES (r)	Mean Difference (I-J)	SD	ES (r)
**DL/EF**	DL/IF	1.01 *	0.07	0.95	1.05 *	0.04	0.98	1.21 *	0.07	0.96
	DL/C	2.02 *	0.07	0.98	2.06 *	0.04	0.99	2.44 *	0.07	0.99
	T/EF	3.05 *	0.07	0.98	3.01 *	0.04	0.99	2.83 *	0.07	0.99
	T/C	3.54 *	0.07	0.99	3.3 *	0.04	0.99	3.8 *	0.07	0.99
**DL/IF**	DL/C	1 *	0.07	0.99	1 *	0.04	0.99	1.23 *	0.07	0.97
	T/EF	2.03 *	0.07	0.98	1.95 *	0.04	0.99	1.62 *	0.07	0.98
	T/C	2.52 *	0.07	0.99	2.24 *	0.04	0.99	2.58 *	0.07	0.99
**DL/C**	T/EF	1.02 *	0.07	0.95	0.95 *	0.04	0.98	0.38 *	0.07	0.74
	T/C	1.52 *	0.07	0.99	1.23 *	0.04	0.99	1.35 *	0.07	0.98
**T/EF**	T/C	0.49 *	0.07	0.83	0.28 *	0.04	0.82	0.96 *	0.07	0.96

## Data Availability

The data that support the findings of this study are available from the corresponding author, S.O., upon reasonable request.

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
