# Peer review of "External Focus or Differential Learning: Is There an Additive Effect on Learning a Futsal Goal Kick?"

_ijerph, 2021, doi:10.3390/ijerph19010317_

Round 1
Reviewer 1 Report
This study investigated the effects of differential learning and elements of OPTIMAL theory on learning a goal kicking skill in futsal under aspects of external and internal focus. The topic seems to be interesting for readers. I have a few comments which I hope will be helpful for improving this manuscript.
General comment: The study has the potential to add support to the already published knowledge, however the authors must clearly outline the novelty of their research.
p.2, lines 92-93 – this sentence requires references
p.2-3, lines 97-99 - The hypothesis must be more specific, please clarify the ideas and show the direction of this hypothesis, with regards to the differences. Moreover the hypothesis should be supported by references.
p.3, lines 112-113 Was the sample size really large enough for the statistical tests?. For what kind of statistical tests was the calculation made? Provide some relevant information for readers.
p.7, lines 211-239 – Why did the authors not include the effect size estimation for all comparisons, in the results section? The analysis of effect sizes for the post-hoc tests should be included in this section. Furthermore, indicate how effect sizes were interpreted. The significance of differences should also be marked in Figure 3.
p.7, line 239 – Where in this manuscript is Figure 4 ?
p.8 (Discussion) - The discussion section seems to be too short. I suggest explaining the neurophysiological mechanism underlying motor skills learning. What were the limitations of the study? This section should be expanded to make it more clear and complete.
p. 8-9, lines 286-304 – In my opinion, authors repeated findings in the conclusion section. Please consider formulating more consistent and factual conclusions.
The findings should be elaborate and the conclusion should be improved.
Author Response
We would like to thank the editor and reviewers for their valuable comments, they have improved the quality of our study. We diligently addressed the comments and look forward to clarifying further as needed. Manuscript changes were highlighted in red.
Response to Reviewer 1
This study investigated the effects of differential learning and elements of OPTIMAL theory on learning a goal kicking skill in futsal under aspects of external and internal focus. The topic seems to be interesting for readers. I have a few comments which I hope will be helpful for improving this manuscript.
General comment: The study has the potential to add support to the already published knowledge, however the authors must clearly outline the novelty of their research.
We are glad that you found this article interesting.
p.2, lines 92-93 – this sentence requires references
Reference 44 added - Now line 103
p.2-3, lines 97-99 - The hypothesis must be more specific, please clarify the ideas and show the direction of this hypothesis, with regards to the differences. Moreover, the hypothesis should be supported by references.
Thank you so much. The hypothesis was modified based on your comment. Now Lines 104-110.
“Therefore, the general aim of this study was to evaluate the effects of the DL approach and elements of the OPTIMAL-theory on goal-kicking performance in futsal in comparison to a purely repetitive approach. More specifically, the hypotheses to be tested were that the DL intervention would outperform the external-focus intervention, but that both interventions would have significantly larger effects compared to repetition-only learning, and finally, that the combination of both interventions would outperform the DL-only intervention without or with internal attentional focus [35, 43].”
p.3, lines 112-113 Was the sample size really large enough for the statistical tests?. For what kind of statistical tests was the calculation made? Provide some relevant information for readers.
The required description was added to the relevant section. – Now lines 122-125.
“Using G-Power, the sample size was estimated at the significance level of 0.05, with a 0.80 statistical power and effect size of 0.6 (medium to large effect size), and the statistical method of One-way analysis of variance [44-45]. According to this 40 people were assigned into five groups.”
p.7, lines 211-239 – Why did the authors not include the effect size estimation for all comparisons, in the results section? The analysis of effect sizes for the post-hoc tests should be included in this section. Furthermore, indicate how effect sizes were interpreted. The significance of differences should also be marked in Figure 3.
The effect size and interpretation of them, were reported for the post-hoc tests, and also some explanation was added to the legend of figure 3.
p.7, line 239 – Where in this manuscript is Figure 4 ?
Sorry for the inconvenience, there is no figure 4. This should have been Table 4 instead and is replaced in the manuscript.
p.8 (Discussion) - The discussion section seems to be too short. I suggest explaining the neurophysiological mechanism underlying motor skills learning. What were the limitations of the study? This section should be expanded to make it more clear and complete.
The whole discussion and conclusion were extended by a neurophysiological discussion section and repetitions were removed. Lines 284-288 and 302-351.
- 8-9, lines 286-304 – In my opinion, authors repeated findings in the conclusion section. Please consider formulating more consistent and factual conclusions. The findings should be elaborate and the conclusion should be improved.
The whole discussion and conclusions were extended and rewritten. Lines 284-288 and 302-351.

Reviewer 2 Report
I have attached my review

Author Response
Dear reviewer 2
thank you very much for your reviewing comments. The comments will help to improve the quality of the manuscript substantially. Our detailed responses you can see in the following - they are all related to a modified manuscript that is uploaded in the attachment.
Kind regards
This is an interesting study that investigates a critical component of coaching – the relationship between the method of relaying information to athletes and performance outcome. Also, the examination of retention following respective methods is paramount and will certainly add to the body of data within this field. However, the manuscript requires significant revision.
Be sure spelling, punctuation and notations are all correct and identical throughout the manuscript. Also, check grammar throughout the manuscript.
Throughout the text, the grammar was re-examined and the necessary corrections made
Table 1: Change “STD” to more appropriate “SD”. Also, be sure in-text spacing is uniform.
Was replaced
Table 2: What was the time period difference between the retention and transfer tests? Presently, it reads as though the transfer test occurred immediately after the retention test.
Yes, the transfer-test was conducted immediately after the retention test. This information is now added in table 2 (last column)
Figure 2: Omit “following” as this is understood; same in Table 3.
Done.
When investigating motor learning why did the authors elect to employ blocked order versus random order?
The aim of our study was a comparison of the effects of elements of OPTIMAL training approach with the differential Learning approach. The OPTIMAL training approach is so far mainly applied in the context of the traditional repetitive training where exercises are repeated until a certain level of perfection before the next exercise is practiced. This approach can be considered as being blocked. A comparison of blocked and random approaches is rather a matter in connection with questions related to Contextual interference learning. A comparison of blocked and random would be associated with a fixed number of different exercises that has to be practiced in one group in a blocked manner and in the other group in a random but still repetitive sequence. Differential learning in the present form has no repetitions and therefore no fixed number of exercises but rather a chaotic characteristic. Therefore, the term “blocked” is only used in the context of the test, in which the shooting conditions mainly differed in their relative position to the target. All the test exercises were not specific subject of the intervention protocol as it is in the typical CI experiments. A comparison of the repetitive with the chaotic intervention by means of a blocked test therefor seems plausible for this first step. Because former studies and the theory of DL already gave indication for better transfer performance in comparison to repetitive learning the focus in this study was rather on the external versus internal focus than on the sequence. Nonetheless, this question would be interesting to solve in future studies.
Summing up:
The following two sentences were added in line 341-343:
“Another interesting question is the influence of the test sequence. Future research would have to clarify to what extent the interventions have different influences on blocked or random test order.”
Who administered the verbal instruction and conducted respective practices?
All training sessions and training instructions were led by one of the authors of this article, who was also a futsal expert. See line 214-215.
Statistical Analysis
What metrics were used for analysis – best, average, median?
We used the average scores obtained in 35 trials as the final score of each person in the shooting test. This was added to the text
Also, the correct notation for Cohen’s effect size is “d”; r signifies Pearson’s correlation coefficient. Please revise.
The r values can be also expressed in d values by means of the equation that is given in Cohen (cf. Ref. 49) or on https://www.psychometrica.de/effect_size.html - :~:text=If%20the%20two%20groups%20have,of%20their%20common%20standard%20deviation
We added the recalculated d values. Lines 218-224.
Discussion
Please give an honest assessment of the limitations to the current study.
The limitations have been rewritten according to all reviewers. See Lines 329 -345
Please see below for specific comments. All the Best.
Comments
Page 1, Lines 29 – 33: Although the message is there, be more concise to enhance clarity.
Done.
Page 1, lines 37 – 38: Replace “mainly” with “main” for readability. [14]?
Was replaced. Line 45
Page 2, line 62: Please spell-out “DL” before using abbreviation.
Done. line 69
Page 2, lines 63 – 64: Check parentheses.
Thank you. Checked and corrected. Lines 70-71
Pages 2, lines 66 - 68: Please revise for clarity.
Was revised. Lines 62 – 68.
Page 2, line 69: Either place “furthermore” at the beginning of the sentence, or omit it.
the sentence has been rephrased. Lines 74 -76.
Page 2, lines 69 – 70: This statement appears random. How does this experiment fit with case being built by the authors? Please revise.
The original sentence:
“In a volleyball experiment, furthermore, the players had to play while moving like various animals that were creatively announced by each new server “
Was rewritten:
In a volleyball experiment the DL-idea of increased noise was realized by means of moving like various animals (metaphorically) that were creatively announced by each server.
Page 2, lines 77 – 78: What is meant by “in the beginning”?
The sentence was rephrased in the following way:
Originally:
While the DL approach in the beginning focused more on the variety of stimuli that result from an interaction of active and passive forces and conditions applied on the organization of central nervous structures through variable movements via mechano-receptors, the OPTIMAL
Now:
While the DL approach in the beginning focused more on the variety of external stimuli that result from an interaction of active and passive forces and conditions applied on the organization of central nervous structures through variable movements via mechano-receptors (Schöllhorn 2000) before the increased fluctuations were adapted to an individual’s movement noise (Schöllhorn et al 2006), the OPTIMAL …
Page 2, line 81: Replace “and” with “as it”, and omit the comma.
Done. now line 87
Page 2, lines 93 – 94: Replace “as compared with the” with “than an”.
Was replaced. Now Line 100
Page 2, line 95: Omit “has been” for conciseness.
The whole paragraph was rephrased - lines 101-107
Page 2, line 97: Avoid using pronouns (eg. “Our”) in scientific writing. Replace with “The”, or “It was hypothesised…”. Also, since a hypothesis is a prediction the statement should read, “…there will be…”.
The whole paragraph was rephrased - lines 101-107
Page 3, line 98: What is meant by “external tradition” and “traditional control”?
The whole paragraph was rephrased - lines 101-107
Page 3, lines 104 – 107: For sub-group notations please revise as such, “(DL/EF, n = 8)”. Also, add “e)” for the last sub-group.
Done. now line 114
Page 3, line 107: Omit “to choose participants” as this is understood.
Done. now line 115
Page 3, line 108 and 109: Omit “On the other side” and “Moreover” to reduce wordiness.
Done. Now line 116, 117
Page 3, line 112: Please note what version of G*Power was used in parentheses.
Version 3.1 was added. - now Line 119
Page 3, line 116: Please replace “weight” with “mass” for scientific correctness, as “weight” refers to exerted force and “mass” refers to amount of matter.
Replaced. Now line 125
Page 3, line 117: Do the authors mean “cm” for height?
adjusted. Now line 126
Page 3, line 119: Revise to “was completed”.
Was corrected. Now line 128
Page 3, line 131 and Table 2: Be sure spelling is consistent (e.g. posttest and pretest, respectively).
We checked it and adjusted the text.
Page 4, line 158: This requires a brief explanation and referencing for readers who wish to replicate the study.
Rephrased - lines 169 – 171.
Page 4, line 159: Check grammar.
Done. Line 171
Page 4, line 160: Do not abbreviate; be consistent with “figure 2”.
Done. Line 172.
Page 6: lines 174 – 175: Be more concise in verbiage.
The sentence
“Each exercise was assigned a number, and then, these have been randomly drawn during the intervention. Up to three of the exercises listed in table 3 were combined.”
Was replaced by
Each exercise was assigned a number. The numbers were written on slips of paper, and all put in a box. Slips of paper with the numbers were then randomly drawn from the box during each intervention session. The assigned exercise was then to be performed by the participants. Up to three of the exercises listed in Table 3 were combined.
Page 6, lines 182 – 192: Replace “oral” with verbal”.
Replaced. Lines 203-223
Page 7, line 209: Report the level of significance as the traditional p-value. Also, omit “Indeed” to reduce wordiness.
Done. Line 232
Figure 3: Why is the metric referred as “Delayed Retention” when it has been consistently?
It was the retention test. The wording was adapted.
Page 7, line 224: Please note the magnitude of the large ES since there is no table.
Done. The table is provided now.
Page 7, lines 224 – 225: Replace “As a further matter” with “Furthermore” to reduce wordiness.
replaced - see line 256
Page 7, lines 225 – 228: Please revise for clarity and readability.
Rephrased - see line 256-60.
Page 7, line 234: Where is Table 4?
Table 4 is included now.
Page 7, lines 229 – 239: Why are effect sizes not noted? Please include in-text where appropriate if a table will not be provided.
Done. The table is provided.
Page 8, line 249: Be more concise to reduce wordiness.
Done. Lines 290-295.
Page 8, lines 250 -251: What is meant by this statement? The concept may require a brief explanation before fully discussing.
The sentence was rephrased for better clarity. – line 291-292
Page 8, lines 251 – 253: Please revise for clarity and conciseness.
Rephrased – see 292-296
Page 8, line 253: Again, what is meant by “DL by time”?
Rephrased – see 291-296
Page 8, lines 260 – 262: This is an interesting and potentially important concept. Please elaborate a bit more to give context, the fit the current findings into the context set.
More details added
Page 8, line 266: Replace “go along” with “agree”. And revise this sentence for conciseness.
Done. The whole paragraph has been rephrased - line 312-324
Page 8, lines 276 – 277: Please revise for clarity.
Done. The whole paragraph has been rephrased - line 312-324
Page 8, lines 278 – 283: Break this statement up into separate sentences to avoid a run-on.
Done. The whole paragraph has been rephrased - line 312-324

Reviewer 3 Report
- The manuscript titled "External Focus or Differential Learning: Or is there an additive effect on learning a futsal goal kick? " is quite fascinating to read. The research was well conceived, executed and the findings well reported and discussed.
- However, the abstract, lines 9-11, should be edited to read thus: "Background: How to optimally promote the process of acquiring and learning a new motor skill is still one of the basic questions often raised in training and movement science, rehabilitation, and physical education. (2) Methods: This study aimed at investigating the effects of differential learning (DL) and elements of OPTIMAL theory on learning a goal kicking skill in futsal, especially under aspects of external and internal focus."
- Page 2, line 56, should have "the question is arisen" edited to read. "the question arises about the necessity"...the whole sentence may need rephrasing.
- Page 2. line 95 should be revised thus, "Therefore, this study aimed at investigating"
- Page 3, line 103, delete "have" voluntarily.....and on line 110, replace "study 'has been' with 'was'
- Page 3. line 119, replace "would be" with 'were' and on line 128, delete "also"
- Page 8, Lines 275-277, the sentence needs editing for clarity: " However, the practical interventions derived from OPTIMAL-theory seem to rep- resent an add-on that could be of major interest when it comes to the absolute limits of performance should"
- Similarly, this sentence on page 8, lines 278-283, could be broken down into shorter sentences and also for the citation to be fitted in appropriately: "The present study results were based firstly on the assumption that DL exercises lead to detecting adaptive solutions through increased fluctuations in the individual, and seem to support the learning ability Hosseini, et al. (2017), and secondly on the finding that DL exercise, albeit at an optimal relation of internal and external noise frequency [61], moves the brain to a more stress-resilient state by down-regulating the frontal lobe activation [36]."
Shorter sentences convey the message more effectively than long winding ones.
Author Response
We would like to thank the editor and reviewers for their valuable comments, they have improved the quality of our study. We diligently addressed the comments and look forward to clarifying further as needed. Manuscript changes were highlighted in red.
Response to Reviewer 2
- The manuscript titled "External Focus or Differential Learning: Or is there an additive effect on learning a futsal goal kick? " is quite fascinating to read. The research was well conceived, executed and the findings well reported and discussed.
Thank you very much for your positive feedback. We have very excited and happy of this positive comment.
- However, the abstract, lines 9-11, should be edited to read thus: "Background: How to optimally promote the process of acquiring and learning a new motor skill is still one of the basic questions often raised in training and movement science, rehabilitation, and physical education. (2) Methods: This study aimed at investigating the effects of differential learning (DL) and elements of OPTIMAL theory on learning a goal kicking skill in futsal, especially under aspects of external and internal focus."
We adapted the abstract according to the suggestion. Now lines 16-20.
- Page 2, line 56, should have "the question is arisen" edited to read. "the question arises about the necessity"...the whole sentence may need rephrasing.
The sentence is rephrased. Now lines 64-69.
- Page 2. line 95 should be revised thus, "Therefore, this study aimed at investigating"
The sentence is rephrased. Now lines 104-110.
- Page 3, line 103, delete "have" voluntarily.....and on line 110, replace "study 'has been' with 'was'
Thank you so much for your good comments. Changes done. See line 114 and Line 121.
- Page 3. line 119, replace "would be" with 'were' and on line 128, delete "also"
Done. See Line 131 and line 141.
- Page 8, Lines 275-277, the sentence needs editing for clarity: " However, the practical interventions derived from OPTIMAL-theory seem to rep- resent an add-on that could be of major interest when it comes to the absolute limits of performance should
The sentence was edited. The whole discussion and conclusion has been extended and rewritten. Lines 284 - 351
- Similarly, this sentence on page 8, lines 278-283, could be broken down into shorter sentences and also for the citation to be fitted in appropriately: "The present study results were based firstly on the assumption that DL exercises lead to detecting adaptive solutions through increased fluctuations in the individual, and seem to support the learning ability Hosseini, et al. (2017), and secondly on the finding that DL exercise, albeit at an optimal relation of internal and external noise frequency [61], moves the brain to a more stress-resilient state by down-regulating the frontal lobe activation [36]." Shorter sentences convey the message more effectively than long winding ones.
Thanks for your very clever comment. The sentence was divided into two sentences. See lines 321-325.

Round 2
Reviewer 1 Report
I accept the answers and have no further comments.
Author Response
Thank you very much for all your effort.